# Characterization of Global DNA Methylation in Different Gene Regions Reveals Candidate Biomarkers in Pigs with High and Low Levels of Boar Taint

**DOI:** 10.3390/vetsci7020077

**Published:** 2020-06-13

**Authors:** Xiao Wang, Haja N. Kadarmideen

**Affiliations:** Quantitative Genomics, Bioinformatics and Computational Biology Group, Department of Applied Mathematics and Computer Science, Technical University of Denmark, 2800 Kongens Lyngby, Denmark; xiwa@dtu.dk

**Keywords:** DNA methylation, promoter, exon, intron, differentially methylated region, boar taint

## Abstract

DNA methylation of different gene components, including different exons and introns, or different lengths of exons and introns is associated with differences in gene expression. To investigate the methylation of porcine gene components associated with the boar taint (BT) trait, this study used reduced representation bisulfite sequencing (RRBS) data from nine porcine testis samples in three BT groups (low, medium and high BT). The results showed that the methylation levels of the first exons and first introns were lower than those of the other exons and introns. The first exons/introns of CpG island regions had even lower levels of methylation. A total of 123 differentially methylated promoters (DMPs), 194 differentially methylated exons (DMEs) and 402 differentially methylated introns (DMIs) were identified, of which 80 DMPs (DMP-CpGis), 112 DMEs (DME-CpGis) and 166 DMIs (DMI-CpGis) were discovered in CpG islands. Importantly, *GPX1* contained one each of DMP, DME, DMI, DMP-CpGi, DME-CpGi and DMI-CpGi. Gene-GO term relationships and pathways analysis showed DMP-CpGi-related genes are mainly involved in methylation-related biological functions. In addition, gene–gene interaction networks consisted of nodes that were hypo-methylated *GPX1*, hypo-methylated *APP*, hypo-methylated *ATOX1*, hyper-methylated *ADRB2*, hyper-methylated *RPS6KA1* and hyper-methylated *PNMT*. They could be used as candidate biomarkers for reducing boar taint in pigs, after further validation in large cohorts.

## 1. Introduction

In approximately 80% of non-castrated male pigs, the accumulation of skatole [1] and androstenone [2] in backfat causes offensive boar taint (BT) flavour. Approximately 75% of consumers in European countries were sensitive to BT in porcine meat [3,4,5].

DNA methylation is a major epigenetic mechanism that directly causes a chemical modification to DNA [6] conferred by the covalent transfer of a methyl group to the C-5 position of cytosine [7]. Since being discovered 40 years ago, methylation has been established as a major epigenetic factor influencing gene activities [8].

DNA methylation in the promoter regions is associated with transcriptional repression at the level of gene regulation [9,10], likely because DNA methylation tends to reinforce chromatin-based silencing [11]. However, to silence genes at transcription, DNA methylation at high cytosine and guanine dinucleotide (CpG)-density promoters is not necessary [12,13]. In fact, unmethylated CpG islands are found in more than one-half of the gene promoter regions, but they are not associated with the gene transcriptional activity [13].

Generally, splicing patterns are influenced by changes in intragenic DNA methylation; for example, the methylation pattern of alternatively spliced exons changes and subsequently alters the exon inclusion levels positively during the construction of splice variants [14,15]. Different methylation levels have been observed in different exons and in exons of different lengths [16]. The findings from a genome-scale intragenic methylation analysis revealed negative strong correlations between gene expression and DNA methylation in first exons but weak correlations in other exons and introns [17]. Therefore, first exon methylation was found to be closely associated with transcriptional silencing [18]. Additionally, Bieberstein et al. (2012) [19] found that the length of the first exon has a regulatory effect on transcription and that it is useful for the prediction of gene activity.

Endogenous introns are important for gene expression, as removing them dramatically reduces transcription and adding them markedly increases transcription [20,21,22,23,24]. Kim et al. (2018) [25] suggested that there is an inverse association of intron methylation and expression, so hypo-methylation of the first intron is correlated with high gene expression levels [26]. Nevertheless, when first introns with high levels of methylation are also located in CpG islands, the gene (e.g., *EGR2*) is expressed at a high level [27].

We have previously published descriptions of the epigenetic characteristics of the porcine genome and its relationship with boar taint level [28,29]. However, a detailed DNA-methylation characterization of different gene regions was not reported in this previous study, despite the important relation of different levels of differentially methylated regions (DMRs) in gene components (e.g., promoters, exons and introns) to BT level. Therefore, the objective of this study was to investigate the methylation of porcine reference gene components (i.e., promoters, different exons and different introns) for BT traits and to identify the related DMRs of gene components. Our previous study revealed 32 significant gene differentially methylated cytosines (DMCs) associated with the BT trait [28]; however, using the same materials used in the first study for this study, we focused on the DMRs of intragenic regions, especially the promoter, first exon and first intron regions, as a single entity. In addition, the interactions of methylated CpG island with promoters, exons and introns were investigated to determine the influences of CpG islands on intragenic methylation.

## 2. Materials and Methods

### 2.1. RRBS and Animal Data

This study was conducted using reduced representation bisulfite sequencing (RRBS) sequencing data from nine porcine testis tissue samples that are accessible through Gene Expression Omnibus (GEO) (https://www.ncbi.nlm.nih.gov/geo/query/acc.cgi?acc=GSE129385). Only cytosines in the cytosine and guanine dinucleotide (CpG) sites were used in our study. Quality control (i.e., removal of adapter and short reads) and alignment of RRBS reads were performed using *Trimmomatic* software [30] and *Bismark* software [31], respectively, based on the same parameters used in our previous study [28,29]. Furthermore, the unqualified read coverages of cytosines (i.e., ≤10 counts and ≥99.9th percentile) were also trimmed following previous studies [28,29,32,33]. Based on the publicly available information of experimental animals published in our previous study [28,34], nine testis tissue samples from three BT groups (i.e., low, medium and high BT groups) of pigs were used in this study, so each BT group had three pigs as replicates. Three BT groups for the nine pigs were formed based on the BT estimated breeding values (EBVs) (i.e., sum of the EBVs of skatole concentration and human nose score (HNS)), where pigs of extreme low or high EBVs belonged to low or high BT groups and pigs with EBVs closest to the mean belonged to the medium BT group, which has been presented and described in the previous study [28,34].

### 2.2. Porcine Reference Gene Components

Based on the reference sequence genes (RefSeqGenes) of *Sus scrofa* (http://genome.ucsc.edu/cgi-bin/hgTables), a total of 4473 RefSeqGenes with 4412 promoters, 36,286 exons and 31,912 introns were investigated through annotations using R package *genomation* [35]. Different from our previous study that calculated the differentially methylated cytosines (DMCs) associated with BT trait [28], this study focused on the DMRs of intragenic regions (i.e., promoter, exon and intron regions) as a single entity. In addition, the overlaps of methylated CpG island regions with intragenic regions were also investigated to determine the influences of CpG islands on intragenic methylation. Here, we calculated the methylation levels of the promoter-CpG island/exon-CpG islands/intron-CpG island when cytosines were located in both promoter/exon/intron and CpG island regions, for example, cytosines were located in the promoter-CpG island regions (Figure 1). 

Then, we used R package *GeneDMRs* (https://github.com/xiaowangCN/GeneDMRs) [36] to obtain weighted methylation levels for each BT group, *Q*-values by the false discovery rate (FDR) method [37] and methylation differences (i.e., difference in weighted methylation levels between the low and high BT groups) for all promoters, exons, introns and their regions overlapped with CpG islands. Here, if the methylation difference was positive, the region was defined as hyper-methylated; thus, a negative methylation difference represents hypo-methylation.

The weighted methylation levels were calculated by the methylated reads number divided by the total reads number given the weights following the previous study [28]:(1)$$\overset{n}{\underset{i=1}{\sum }}\frac{\sum_{j=1}^{m}MR_{ij}}{\sum_{j=1}^{m}TR_{ij}}*W_{ij}\;and\;W_{ij}=\frac{\sum_{j=1}^{m}TR_{ij}}{\sum_{i=1}^{n}\sum_{j=1}^{m}TR_{ij}},$$
where $MR_{ij}$ and $TR_{ij}$ are the methylated and total reads number of the involved cytosine site j at a given region of individual i, m is the total number of cytosine sites involved in this region, n is the total individual number of one BT group and $W_{ij}$ is the weight of reads of the involved cytosine site j of individual i.

The statistical analysis was following the previous study [33] in the logistic regression model:(2)$$\ln\left(\frac{\pi_{i}}{1-\pi_{i}}\right)=u+{\beta T}_{i}$$
where $\pi_{i}$ is the weighted methylation levels at the given region, u is the intercept, and $T_{i}$ is the BT group indicator. 

### 2.3. Differentially Methylated Gene Components, Significant Enrichments and Interaction Networks

The differentially methylated promoters (DMPs), differentially methylated exons (DMEs), differentially methylated introns (DMIs), DMP-CpG islands (DMP-CpGis), DME-CpG islands (DME-CpGis) and DMI-CpG islands (DMI-CpGis) were defined when *Q*-values were less than 0.05.

The enrichment of GO terms for DMP-CpGi-related genes was determined with the DAVID website (https://david.ncifcrf.gov/) and visualized by the *GOplot* R package [38]. The pathway enrichments for hypo-methylated and hyper-methylated DMP-CpGi-related genes were performed in *clusterprofiler* R package [39]. Afterwards, The DMP-CpGi-related genes enriched in significant GO terms and pathways were used to create gene–gene networks by the *GeneMANIA* tool [40,41].

## 3. Results

### 3.1. Methylation Levels of Promoters, Different Exons and Different Introns

In this study, the cytosines in CpG sites were found in only 3029 genes with 2295 promoters, 2725 exons and 4349 introns, based on reduced representation bisulfite sequencing (RRBS) data. According to the distribution of all exon and intron positions, the frequency of positions decreased as the number of ordinal positions increased, so the first ordinal position had the most exons and introns. Additionally, exons and introns were in fewer than 20 ordinal positions; therefore, the number of exons and introns in most genes was less than 20 (Appendix A). In fact, the proportions of both the first twenty exons and first twenty introns out of all exons (2590/2725) and introns (4109/4349) were larger than 90% (Figure 2A,C,E). Obviously, the methylation levels of the promoters were lower compared to the methylation levels of the first twenty exons and the first twenty introns (Figure 2). However, exons/introns at different ordinal positions were found to have different methylation levels, but such differences in the three BT groups were very small (Appendix A). The average methylation levels of the first exon (21.64%) and first intron (31.05%) for the three BT groups were lower than the average methylation levels of the other exons and introns (Figure 2A,C,E). However, we investigated the specific genes such as *GPX1* and found that methylation levels of intergenic regions of *GPX1* were higher than intragenic regions, where low BT groups had higher methylations than other two BT groups (Figure 3).

When considering the CpG island overlapped with intragenic regions, cytosines were found to be involved in 2196 genes with 1942 promoters, 1635 exons and 1796 introns. The methylation of the promoters and the first twenty intragenic regions that were exclusively located in CpG islands was generally lower than the methylation of all promoters and the first twenty gene components, particularly the promoters; the 1st, 13th and 14th exons; and the 1st and 20th introns (Figure 2). Obviously, the methylation levels interacting with CpG islands of the 13th and 14th exons were significantly (*p* < 0.001) lower than the same-exon methylations without CpG island interactions (Appendix A).

The methylation levels of the first exons were observed to increase as the lengths increased, but this trend was not as notable for the first introns (Appendix A). In terms of their overlaps with CpG islands, methylated exons remained stable (Appendix A), but methylated introns tended to have greatly decreased variable lengths (Appendix A).

The methylation differences between the low and high BT groups ranged from −40% to +30% for promoters, first exons and first introns in the same genes, and the first exons had greater methylation differences than were found among the promoters and the first introns. Additionally, most methylation differences in the promoters, first exons and first introns were at the zero point, which indicated that the differential methylation levels of the gene components were generally analogous in the low and high BT groups. The highest Pearson correlation coefficient (PCC) for methylation differences was associated with the comparison of promoters and first exons (PCC = 0.69), while the correlation coefficients were lower for comparisons of promoters with first introns and first exons with first introns (Appendix A).

### 3.2. Differentially Methylated Promoters, Exons, Introns and Their Overlaps with CpG Islands

According to the filtering criterion of *Q*-values < 0.05, among the identified 123 differentially methylated promoters (DMPs), 194 differentially methylated exons (DMEs) and 402 differentially methylated introns (DMIs), a total of 80 DMPs (DMP-CpGis), 112 DMEs (DME-CpGis) and 166 DMIs (DMI-CpGis) were discovered in CpG islands. The details of the DMPs, DMEs and DMIs are listed in Appendix A, and the details of DMP-CpGis, DME-CpGis and DMI-CpGis are listed in Appendix A. Manhattan plots of genome-wide DNA methylation in promoters, exons and introns for BT are shown in Figure 4.

The number of common genes related to DMPs, DMEs and DMIs was three and two of the three genes were related to DMP-CpGis, DME-CpGis and DMI-CpGis (Figure 5A,B). Among them, *TSPAN9* and *GPX1* contained one each of DMP, DME, DMI, DMP-CpGi, DME-CpGi and DMI-CpGi (Table 1). The percentage of DME, DMI, DME-CpGi and DMI-CpGi in the first and other ordinal positions varied. The first exons constituted a relatively small proportion (7.6%~7.8%) of the DMEs regardless of whether they were in CpG islands, while the first introns constituted a relatively larger proportion (14.7%) of the DMIs when they were excluded from CpG islands (Figure 5C,D). The methylation differences of DMP, DME, DMI, DMP-CpGi, DME-CpGi and DMI-CpGi were close to zero (Figure 5E,F).

Finally, 7 hypo-methylated and 13 hyper-methylated genes were presented as the top 20 DMP-CpGi-related genes. Additionally, 12 hypo-methylated and 8 hyper-methylated DME-CpGi-related genes, together with 13 hypo-methylated and 7 hyper-methylated DMI-CpGi-related genes are also listed in Table 2. The most significant DMP-CpGi-related gene (*Q*-value = 2.8 × 10^−16^), hypo-methylated *POU2AF1*, also had the larger methylation difference (18.4%) than other DMP-CpGi-related genes (Table 2). In the top 20 DME-CpGi-related genes, 5 first DME-CpGis were identified that were associated with *GLI2*, *RTL1*, *ZNF205*, *SOX9* and *HOXA5*, whereas 6 first DMI-CpGis related to *SCD5*, *GPX1*, *PDX1*, *PKD1*, *TBCD* and *HOXA10* were identified in the top 20 DMI-CpGi-related genes. Furthermore, the promoters of *RTL1* and *HOXA5* were tested as DMPs that were in the same methylated directions with their first DME-CpGis (Table 2, Appendix A), thus, hypo-methylated/hyper-methylated promoter-first exon regions of *RTL1*/*HOXA5* could be the entirety for functional methylations.

### 3.3. Biological Enrichment of DMP-CpGi-Related Genes and Interaction Networks

Among the 80 DMP-CpGi-related genes, four correlated with methylation-related biological functions, such as *PNMT* (*Q*-value = 5.9 × 10^−8^ with a methylation difference = −6.2%), *BHMT* (*Q*-value = 3.7 × 10^−5^ with a methylation difference = 2.1%), *BHMT2* (*Q*-value = 5.3 × 10^−3^ with a methylation difference = 1.2%) and *GNMT* (*Q*-value = 2.8 × 10^−2^ with a methylation difference = 1.7%) (Appendix A). The gene–GO term relationship results for 80 DMP-CpGi-related genes also showed that four methylation-function genes were strongly linked to significant GO terms (*P*-value < 0.05), including methylation (GO:0032259) and the S-adenosylmethionine metabolic process (GO:0046500) in the biological process category and betaine-homocysteine S-methyltransferase activity (GO:0047150) and S-adenosylmethionine-homocysteine S-methyltransferase activity (GO:0008898) in the molecular function category. In addition, 13 other genes (*ADRB2*, *APP*, *ATG4D*, *ATOX1*, *DPP4*, *FADD*, *GPX1*, *HNF1B*, *HPSE*, *KCNA5*, *MAL2*, *NTN1* and *PTPRA*) were involved in 11 of the 19 significant GO terms (Figure 6A). In the pathway enrichment analysis, seven genes (i.e., *ADRA1A*, *ADRB2*, *BHMT*, *BHMT2*, *GNMT*, *PPP1CB* and *RPS6KA1*) were enriched in six significant pathways (p.adjust < 0.05) including the adrenergic signaling in cardiomyocytes (ssc04261), the glycine, serine and threonine metabolism (ssc00260), the cGMP-PKG signaling pathway (ssc04022), the cysteine and methionine metabolism (ssc00270), the long-term potentiation (ssc04720) and the salivary secretion (ssc04970) (Figure 6B). In the previous study, 32 DMC-related candidate genes (e.g., *FASN* and *PEMT*) were associated with BT, and these DMCs were located only in the gene components [28]. Here, we summarize the DMIs in the 27th intron of *FASN* and in the 2nd and 3rd introns of *PEMT* (Appendix A). Based on the 20 genes involved in the significant GO terms and pathways, gene–gene networks were created to connect candidate genes (i.e., *FASN* and *PEMT*) for the BT trait. *FASN* is connected to *APLP2*, *PTGR2* and *OXSM*, while *PEMT* is linked to *ADRA1A*, *PTPRA* and *NNMT*. Moreover, *GPX1* was in relationship with a group of genes that included *APP*, *ATG4A*, *ATOX1*, *ADRB2*, *CCS*, *PNMT*, *RPS6KA1* and *NNMT* (Figure 6C).

## 4. Discussion

### 4.1. Methylation Status of Promoters, Different Exons and Different Introns for BT

Most studies have suggested that the DNA methylation levels of introns are lower than those of neighbouring exons in humans and honey bees [42,43]. However, methylation of exons with CpG islands was at a higher level than the methylation of flanking introns in embryonic stem cells and foetal fibroblasts of humans [44]. Our previous results implied that exons in CpG islands had lower methylation levels than the methylations of the intron and other CpG island regions in porcine testis tissue [29]. The findings of this study indicated higher methylation levels of introns at different ordinal positions (Figure 2 and Appendix A). Additionally, both our previous study [29] and Chen’s study [45] suggested that promoters with lower methylation patterns were specific to adult porcine pig testis during spermatogenic cell development. Therefore, such promoter methylation patterns and inversely different methylation trends between exons and introns could be caused by tissue-specific and developmental-specific methylation patterns.

Song et al. (2017) [16] revealed different ordinal positions of exons with different methylation, showing that the first exons had the lowest methylation levels, a finding consistent with this study (Figure 2). Furthermore, another study demonstrated the correlations of different methylation levels in different gene positions with the biological features of exons [46]. Such exon-specific methylation levels were considered to be associated with alternative splicing events [14,15,43]; for example, the expression levels of the exons were negatively related to the DNA methylation levels of the first exons [46]. Decreasing methylation levels of the first exons (Figure 2A,C,E), especially the obvious methylation decrease in the first exons in the CpG islands (Figure 2B,D,F), might be correlated with both promoters and CpG islands. Our results revealed a good relationship (0.689) between promoters and first exons (Appendix A), so first exons can potentially be viewed as integral to the promoter regions [47,48].

It has been reported that the lengths of the exons are strongly linked to exon methylation level [16]; thus, longer genes with many long exons may have higher methylation levels [49]. Based on the length distribution results in this study, methylation levels increased as exon length increased, regardless of whether or not the exons were in CpG islands (Appendix A). However, the interaction with CpG islands resulted in the reduced methylation of introns compared to the initial levels (Appendix A). Since long introns and high intron variability could cause variable methylation levels in the first introns (Appendix A), CpG islands could play a role in reducing the methylation levels of the first introns, according to their different lengths [26].

Anastasiadi et al. (2018) [26] also indicated that the number of DMRs in the first introns was greater than the number of DMRs in the promoters or first exons. Our study found that 10.9%~14.7% of DMIs were in the first introns (Figure 5C,D), which means that the DMRs in the first introns constituted a large proportion of DMIs and all DMRs because the number of DMIs (n = 166~402) was higher than the number of DMPs (n = 80~123) or DMEs (n = 112~194) (Figure 5A,B).

### 4.2. Biological Functions of DMP-CpGis-Related Genes

As reducing methylation in the promoters causes the enhancement of gene expression, the expression levels are negatively correlated with promoter methylation [9]. Generally, promoters in CpG islands are unmethylated, while promoters outside CpG islands are predominantly methylated [13]. Our study showed similar results: reduced methylation levels were observed in the CpG islands of promoters (Figure 2). Therefore, Weber et al. (2007) [13] proposed that methylation is occasionally not necessary in CpG island promoters, as most of them are in hypo-methylated states, but the existing methylation is enough to inactivate promoter-CpG island regions. Among the 80 DMP-CpGi-related genes, 30 DMP-CpGis were in a hyper-methylated state (Appendix A), and 13 of these were involved in the top 20 DMP-CpGis (Table 2).

The significant GO terms identified through this study mainly focused on biological functions of methylation (Figure 6A) and were connected with three hyper-methylated DMP-CpGi-related genes (i.e., *BHMT*, *BHMT2* and *GNMT*) and one hypo-methylated DMP-CpGi-related gene (i.e., *PNMT*) (Table 2). The gene expression patterns of the *BHMT* and *BHMT-2* genes are similar between pigs and humans [50], with *BHMT* being active in porcine pancreas, kidney and liver cortex [51]. *GNMT*, a key component that regulates S-adenosyl-methionine (SAM) catabolism, suppresses the increment of SAM to extend lifespan [52], whereas *PNMT* expression contributes to the mesodermal origin of adrenergic heart cells [53]. In fact, the hyper-methylated genes *BHMT*, *BHMT2* and *GNMT* were also enriched in the significant pathways, where the other involved DMP-CpGi-related genes *ADRA1A*, *ADRB2*, *PPP1CB* and *RPS6KA1* were also hyper-methylated (Figure 6B). Therefore, the findings of GO terms and pathways could provide a functional understanding of promoter methylation in CpG islands.

Based on gene–gene network results, hyper-methylated genes *ADRA1A* and *PTPRA* linking to *PEMT* were involved in three significant pathways (i.e., the adrenergic signaling in cardiomyocytes (ssc04261), the cGMP-PKG signaling pathway (ssc04022) and salivary secretion (ssc04970)) and one significant GO term (i.e., the protein phosphorylation (GO:0006468)). *PTPRA* was reported to promote the cell cycle progression and lead to poor prognosis in squamous cell lung cancer [54]. In addition, the hypo-methylated *GPX1*, one of the DMP, DME, DMI, DMP-CpGi, DME-CpGi and DMI-CpGi related genes, was connected to hypo-methylated *APP*, hypo-methylated *ATOX1*, hyper-methylated *ADRB2*, hyper-methylated *RPS6KA1* and hyper-methylated *PNMT* that were all enriched in significant GO terms and pathways (Figure 6C). In the other studies, *GPX1* was found to be positively correlated with porcine high androstenone and a high activity against lipid peroxidation based on the glutathione metabolism pathway, so they suggested that the interaction partners (e.g., *GST* families) with *GPX1* could be candidate biomarkers in testicular steroid and androstenone biosynthesis of pigs [55].

### 4.3. Implications

To avoid BT odor in porcine meat, selecting low genetic merit of BT with considerable heritability can be effective and efficient [56,57], as surgical castration has implications for animal welfare and results in decreased meat production [58,59]. Currently, multi-omics data analysis was performed to understand complex traits based on systems genomics approaches [60,61,62], in which the epigenome interacted with the genome to affect the transcriptome and subsequent modules such as the proteome and metabolome to a different extent in the design of omics studies [63]. In addition to genomics studies finding significant genetic variants [64,65,66,67] and transcriptomics studies finding differentially expressed genes [68,69,70] associated with BT, an epigenomics study revealed DMCs to decipher the epigenetic regulatory mechanisms of BT [28]. However, this study identified differentially methylated gene components, for example, DMP, first DME and first DMI, that were key components for DNA methylations to regulate gene expressions [9,10,18,19,20,21,22,23,24]. Thus, the DMRs in gene components are valuably potential epigenetic biomarkers for BT, especially for promoter and first DME showing the similar methylation status [47,48]. Our study used relatively small sample sizes, so it is hard to achieve good quality biomarkers, but we will collect new testis tissue samples in another pig population of the same breed to validate our results in vitro studies by performing additional experiments with the methylated and unmethylated CpGs, and quantitative PCR analysis on target genes.

## 5. Conclusions

This study investigated the methylation of porcine gene components (i.e., promoters, different exons and different introns) in BT trait-related genes and identified the related DMRs of the gene components. The results show that the methylation levels of the first exons and first introns were lower than those of the other exons and introns. Moreover, the methylation levels of the first exon/intron CpG islands were even lower. Additionally, as the first exon lengths become longer, their methylation levels increased. According to the differentially methylated analysis, the DMPs/DMEs/DMIs and their overlaps with CpG islands were identified. Analysis of the gene–GO term relationships and pathways of 80 DMP-CpGi-related genes revealed that these genes enriched in significant GO terms and pathways were mainly involved in methylation-related biological functions. The finding that decreasing methylation levels of the first exons were in a strong relationship with promoters, particularly through interactions with CpG islands, caught our attention because they could be considered as a promoter-first exon entity that regulates biological events. Based on the gene–gene network results, hypo-methylated *GPX1* linking *APP*, *ATOX1*, *ADRB2*, *RPS6KA1* and *PNMT* could be used as potential candidate biomarkers for boar taint in pigs after further validation in large populations of the same breed.

## Figures and Tables

**Figure 1 vetsci-07-00077-f001:**
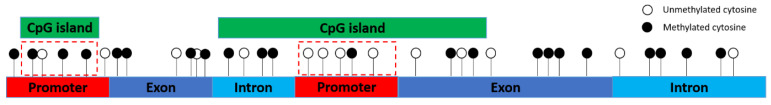
Cytosines (red dotted rectangle) in the promoter-CpG island regions.

**Figure 2 vetsci-07-00077-f002:**
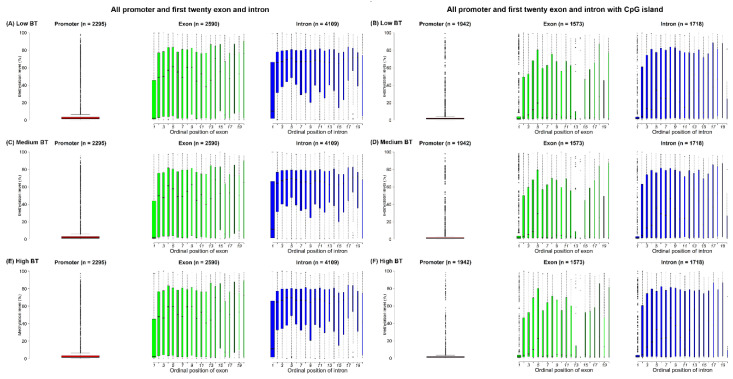
Boxplots of methylation levels (%) of promoters, the first twenty ordinal positions of exons and introns for the (**A** and **B**) low, (**C** and **D**) medium and (**E** and **F**) high BT groups. Note: The widths of boxplots indicate the proportion to the square-root of the number of observations. The number within brackets indicate the number of promoters, exons and introns used in the figure visualization.

**Figure 3 vetsci-07-00077-f003:**
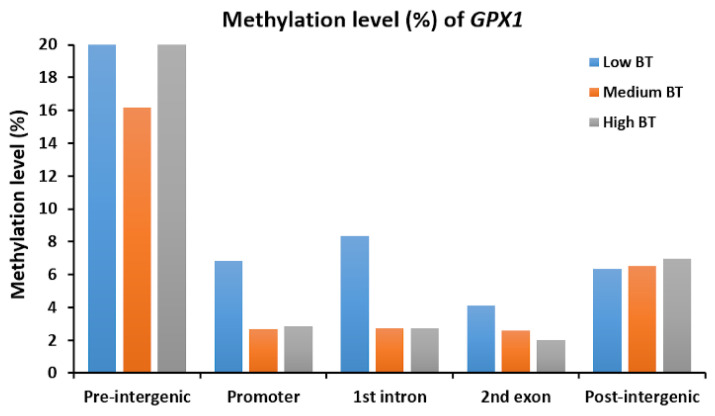
Methylation levels (%) of *GPX1* in different gene components for low, medium and high BT groups.

**Figure 4 vetsci-07-00077-f004:**
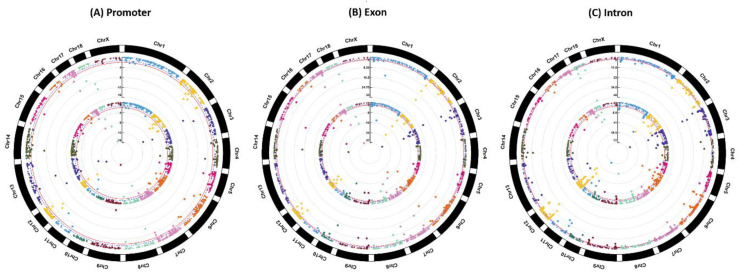
Manhattan plots of genome-wide DNA methylation for BT. Note: Plots from outside track to inside track indicate the *Q*-values of (**A**) promoters, (**B**) exons and (**C**) introns and the *Q*-values of (**A**) promoters, (**B**) exons and (**C**) introns overlapped with CpG islands, respectively. Blue dotted and red solid lines indicate the *Q*-value threshold of 0.05 and 0.01, respectively.

**Figure 5 vetsci-07-00077-f005:**
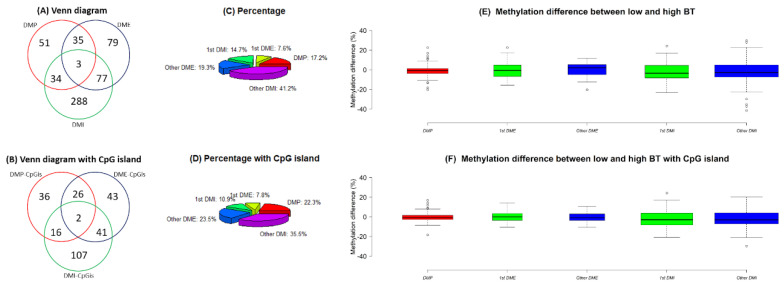
(**A**) Venn diagrams and (**B**) Venn diagrams with CpG islands of all differentially methylated promoters (DMPs), all differentially methylated exons (DMEs) and differentially methylated introns (DMIs). (**C**) Pie charts and (**D**) pie charts with CpG islands of DMPs, and DMEs and DMIs in first and other ordinal positions. (**E**) Boxplots and (**F**) boxplots with CpG islands of methylation differences (%) for DMPs, and DMEs and DMIs in first and other ordinal positions. Note: Methylation difference (%) refers to the difference in methylation levels between the low and high BT groups. The widths of boxplots indicate the proportion to the square root of the number of observations.

**Figure 6 vetsci-07-00077-f006:**
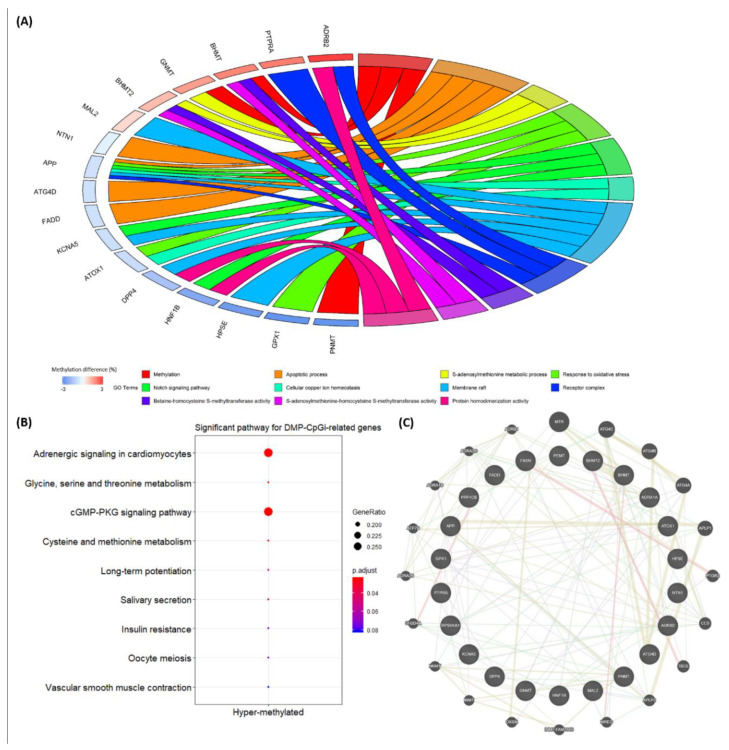
(**A**) Circo plots of relationships between 80 DMP-CpGi-related genes and 19 significant GO terms (*P*-value < 0.05). Note: Methylation difference (%) refers to the difference in methylation levels of DMP-CpGis between the low and high BT groups. (**B**) Dot plots for significant pathways (p.adjust < 0.05). (**C**) Gene–gene networks.

**Table 1 vetsci-07-00077-t001:** Common annotated genes of DMP, DME, DMI, DMP-CpGi, DME-CpGi and DMI-CpGi.

Gene	Chromosome	Gene Description	Region (*Q*-Value & Methylation Difference)					
*TSPAN9*	5	Tetraspanin 9	DMP (1.4 × 10^−2^ & 1.2%)	1st DME (1.5 × 10^−2^ & −8.0%)	6th DMI (1.7 × 10^−10^ & −5.5%)	DMP-CpGi (1.8 × 10^−2^ & 1.2%)	1st DME-CpGi (1.9 × 10^−2^ & −8.0%)	6th DMI-CpGi (2.5 × 10^−5^ & −12.8%)
*MASP2*	6	Mannan binding lectin serine peptidase 2	DMP (4.7 × 10^−8^ & −6.3%)	10th DME (4.4 × 10^−2^ & −5.1%)	9th DMI (5.8 × 10^−5^ & −8.5%)	NA	NA	NA
*GPX1*	13	Glutathione peroxidase 1	DMP (6.5 × 10^−14^ & −4.0%)	2nd DME (2.5 × 10^−2^ & −2.1%)	1st DMI (2.0 × 10^−11^ & −5.6%)	DMP-CpGi (6.6 × 10^−14^ & −4.0%)	2nd DME-CpGi (3.1 × 10^−2^ & −2.1%)	1st DMI-CpGi (2.1 × 10^−11^ & −5.6%)

Note: NA indicates not available. Methylation difference (%) refers to the difference in methylation levels between the low and high BT groups.

**Table 2 vetsci-07-00077-t002:** Top 20 DMP-CpGi-related, DME-CpGi-related and DMI-CpGi-related genes.

Gene	Chromosome	Gene Description	Overlap Region	*Q*-Value	Methylation Difference (%)
*POU2AF1*	9	POU class 2 homeobox- associating factor 1	DMP-CpGi	2.8 × 10^−16^	−18.4
*IGFBP1*	18	Insulin-like growth factor-binding protein 1	DMP-CpGi	5.7 × 10^−16^	9.1
*GPX1*	13	Glutathione peroxidase 1	DMP-CpGi	6.6 × 10^−14^	−4.0
*AMZ1*	3	Archaelysin family metallopeptidase 1	DMP-CpGi	2.1 × 10^−11^	7.9
*SLC7A14*	13	Solute carrier family 7 member 14	DMP-CpGi	5.2 × 10^−11^	2.1
*HOXA5*	18	Homeobox A5	DMP-CpGi	1.7 × 10^−9^	3.4
*PTPRA*	17	Protein tyrosine phosphatase receptor type A	DMP-CpGi	4.2 × 10^−9^	2.2
*PNMT*	12	Phenylethanolamine N-methyltransferase	DMP-CpGi	5.9 × 10^−8^	−6.2
*PRM2*	3	Protamine 2	DMP-CpGi	1.1 × 10^−7^	14.0
*SOD3*	8	Superoxide dismutase 3	DMP-CpGi	2.5 × 10^−7^	4.7
*DCT*	11	Dopachrome tautomerase	DMP-CpGi	7.8 × 10^−7^	8.8
*CLEC4G*	2	C-type lectin domain family 4 member G	DMP-CpGi	3.9 × 10^−6^	11.9
*MIR671*	18	MicroRNA mir-671	DMP-CpGi	8.0 × 10^−6^	−3.5
*TDRD10*	4	Tudor domain containing 10	DMP-CpGi	1.1 × 10^−5^	0.9
*LOC100519311*	5	Uncharacterized LOC100519311	DMP-CpGi	1.6 × 10^−5^	4.9
*CDH5*	6	Cadherin 5	DMP-CpGi	3.3 × 10^−5^	−5.2
*BHMT*	2	Betaine--homocysteine S-Methyltransferase	DMP-CpGi	3.7 × 10^−5^	2.1
*TCTEX1D4*	6	Tctex1 domain containing 4	DMP-CpGi	7.4 × 10^−5^	−6.1
*OXT*	17	Oxytocin/neurophysin I prepropeptide	DMP-CpGi	1.6 × 10^−4^	−5.6
*ADRB2*	2	Adrenoceptor beta 2	DMP-CpGi	1.9 × 10^−4^	3.3
*ZNF217*	17	Zinc finger protein 217	3rd DME-CpGi	8.0 × 10^−32^	−8.2
*AMZ1*	3	Archaelysin family Metallopeptidase 1	7th DME-CpGi	2.7 × 10^−30^	−7.6
*YDJC*	14	YdjC chitooligosaccharide Deacetylase homolog	5th DME-CpGi	6.9 × 10^−24^	24.2
*CHRM1*	2	Cholinergic receptor muscarinic 1	5th DME-CpGi	8.8 × 10^−22^	−8.2
*GLI2*	15	GLI family zinc finger 2	1st DME-CpGi	5.2 × 10^−21^	−10.4
*LRP8*	6	LDL receptor related protein 8	5th DME-CpGi	1.6 × 10^−20^	−13.9
*TNXB*	7	Tenascin XB	48th DME-CpGi	6.4 × 10^−17^	4.3
*IGFBP1*	18	Insulin like growth factor binding protein 1	4th DME-CpGi	5.7 × 10^−16^	9.1
*APOE*	6	Apolipoprotein E	4th DME-CpGi	7.2 × 10^−15^	−12.7
*CAPN2*	10	Calpain 2	6th DME-CpGi	1.1 × 10^−14^	−11.8
*FOXO3*	1	Forkhead box O3	2nd DME-CpGi	1.5 × 10^−14^	−9.1
*RTL1*	7	Retrotransposon Gag like 1	1st DME-CpGi	2.3 × 10^−14^	−4.2
*ADRA1D*	17	Adrenoceptor alpha 1D	3rd DME-CpGi	7.5 × 10^−14^	4.4
*SIGIRR*	2	Single Ig and TIR domain containing	3rd DME-CpGi	2.7 × 10^−13^	11.7
*ZNF205*	3	Zinc finger protein 205	1st DME-CpGi	6.6 × 10^−12^	9.8
*SOX9*	12	SRY-box transcription factor 9	1st DME-CpGi	5.4 × 10^−10^	2.9
*HOXA5*	18	Homeobox A5	1st DME-CpGi	5.5 × 10^−10^	3.2
*COX10*	12	COX10 homolog, cytochrome c oxidase assembly protein, heme A: farnesyltransferase (yeast)	7th DME-CpGi	8.5 × 10^−10^	−9.8
*KLF3*	8	Kruppel like factor 3	3rd DME-CpGi	1.2 × 10^−9^	−9.4
*MYO7A*	9	Myosin VIIA	16th DME-CpGi	1.9 × 10^−9^	−10.9
*CRYL1*	11	Crystallin lambda 1	6th DMI-CpGi	2.7 × 10^−22^	−7.6
*AUTS2*	3	Activator of transcription and developmental regulator AUTS2	5th DMI-CpGi	5.2 × 10^−21^	3.9
*SCD5*	8	stearoyl-CoA desaturase 5	1st DMI-CpGi	2.7 × 10^−17^	−9.8
*SREBF1*	12	Sterol regulatory element binding transcription factor 1	18th DMI-CpGi	1.5 × 10^−14^	−5.1
*ABO*	1	ABO, alpha 1-3-N-acetylgalactosaminyltransferase and alpha 1-3-galactosyltransferase	2nd DMI-CpGi	6.6 × 10^−14^	−4.7
*BANP*	6	BTG3 associated nuclear protein	13th DMI-CpGi	3.0 × 10^−13^	4.4
*PEMT*	12	Phosphatidylethanolamine N-methyltransferase	2nd DMI-CpGi	3.6 × 10^−13^	8.7
*CTSD*	2	Cathepsin D	5th DMI-CpGi	3.6 × 10^−13^	−9.5
*TBCD*	12	Tubulin folding cofactor D	25th DMI-CpGi	2.0 × 10^−12^	−10.0
*GPX1*	13	Glutathione peroxidase 1	1st DMI-CpGi	2.1 × 10^−11^	−5.6
*SLC7A14*	13	Solute carrier family 7 member 14	7th DMI-CpGi	5.2 × 10^−11^	2.1
*PDX1*	11	Pancreatic and duodenal homeobox 1	1st DMI-CpGi	2.9 × 10^−10^	−10.5
*PKD1*	3	Polycystin 1, transient receptor potential channel interacting	1st DMI-CpGi	3.3 × 10^−10^	−8.0
*WDR45B*	12	WD repeat domain 45B	3rd DMI-CpGi	6.2 × 10^−10^	−12.2
*TBCD*	12	Tubulin folding cofactor D	24th DMI-CpGi	1.8 × 10^−9^	−20.9
*NOS3*	18	Nitric oxide synthase 3	2nd DMI-CpGi	2.0 × 10^−9^	9.4
*TBCD*	12	Tubulin folding cofactor D	1st DMI-CpGi	2.3 × 10^−9^	−9.3
*BCO1*	6	Beta-carotene oxygenase 1	5th DMI-CpGi	9.0 × 10^−9^	−6.4
*HOXA10*	18	Homeobox A10	1st DMI-CpGi	1.1 × 10^−8^	2.4
*TBCD*	12	Tubulin folding cofactor D	34th DMI-CpGi	2.4 × 10^−7^	9.6

Note: Methylation difference (%) refers to the difference in methylation levels between the low and high BT groups.

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
