# Peer review of "Characterization of Global DNA Methylation in Different Gene Regions Reveals Candidate Biomarkers in Pigs with High and Low Levels of Boar Taint"

_vetsci, 2020, doi:10.3390/vetsci7020077_

Round 1

Reviewer 1 Report

I do not have further comments.

Reviewer 2 Report

Boar taint is an interesting trait, the manuscript did investigation of DNA methylation of testis in three BT groups. They identified some of differentially methylated genes and exons, which probably could be used as the reference of candidate gene selection. The revised manuscript improved result organization and the English writing. 

This manuscript is a resubmission of an earlier submission. The following is a list of the peer review reports and author responses from that submission.

Round 1

Reviewer 1 Report

Comments to the Author,

This study investigated the methylation of porcine gene components associated with the boar taint (BT) trait by reduced representation bisulfite sequencing (RRBS). The author provided new data that would contribute to characterizing the role of gene methylation in the process of reducing boar taint in pigs, and screened out some candidate biomarkers for boar taint in pigs. Thus, the finding is interesting and contains some useful information for the researchers in the closely related field. Moreover, the result is technically sounded and worthy to be published in Veterinary Sciences. Nevertheless, I have several concerns that need to be addressed before its acceptance for publication.

Minor issues:

  1. How are the criteria for the classification of three BT groups (low, medium and high group) in the collection of experimental animals? The author should describe it in brief.
  2. As mentioned in the “conclusion” part, important associations of the first exons and the first introns are showed in the reduction of boat trait (BT) porcine gene components. Decreased methylation levels in the first exons have a strong relationship with promoters. Meanwhile, the first introns among the DMIs account for the largest proportion of all DMRs, could you provide new insights into related field studies based on these results.

  3. “References section” part, please note the correct writing format and unify. (whether the font is bold, whether it is italic, the issue number of the journal, etc.)

Reviewer 2 Report

This paper reports RRBS sequencing of testis tissue (n= 9) from boars with high and low levels of taint. Primary, the CpG levels within regulatory and structural regions of genes were described. This part is not novel, several studies have previously reported such findings. Then, they identified several differentially methylated regions between high and low group, but their biological relevance with the studied phenotype is unclear, nor sufficiently investigated. The authors perfomed an in silico interaction analysis for the candidate genes only.

Major issues:

  1. To exclude the false positive DMRs (can be observed in RRBS) the methylation of the most promissing CpG sites must be validated with an alternative method. The top gene, POU2AF1 showed 18% difference in methylation between groups. This can have biological relevance. However, most of the candidate CpG sites showed small differences. These differences can also be biologically important, but need to validated in homogenous cells extracts.
  2. The funtional tests need to be done. These requires, in vitro studies with the methylated and non methylated CpGs, and quantitative PCR analysis on target gene(s). The current manuscript is descriptive, lacks novelty and considering the low number of animals studied adds less to the existing knowlegde.

Reviewer 3 Report

The manuscript titled "Characterization of global DNA metylation
in different gene regions reveals candidate biomarkers in pigs
with high and low levels of boar taint" describes the attempts
to find the potential associations between the pattern of metylation
of different DNA regions and the boar taint features of meat.
It is clearly and neatly written, although the study was quite
complex. The paper is well and adequatedly illustrated.
The language seems flawless.
I would however suggest to re-edit the abstract due to its
overloading with details that make it difficult to follow.

Reviewer 4 Report

Boar taint is an interesting trait, the authors conducted descriptive study as for genome-wide DNA Methylation with animals of extreme phenotype of boar taint. I noticed that the authors have published two papers with the same dataset. The present study analyzed the data from gene structure, which maybe provide new information, but the following questions should be considered:

  1. The comparison should consider intergenic CpG level as control.
  2. The promoter and first intron have actual regulatory function to gene expression, which may be displayed different extent of methylation. So, the comparisons should do for the promoter, first exon, first intron, other exons, other introns and intergenic regions. Presently, compare different exons does not make sense.
  3. Why you choose 20 exons? did you do the distribution analysis of exon number for all genes? It is very important information. If we don’t know the distribution, the classification of exons has no background.
  4. What is the exact biological meaning for the 13th exon and 14th exon? If the comparison is conducted with one specific gene, maybe it has some interesting indication, however, if the comparison was conducted with many genes, what’s the meaning? Again, we should have the background of exon distribution.
  5. Figure 5, why you distinguish different exons as exon2-5 and exon6-10, exon 10-20? And what’s the exact meaning for this kind of classification?
  6. The organization of introduction should change. It’s better to introduce the background of your biologic question first, boar taint.
  7. The English writing should improve, there are lots of long sentences, and a general problem is that the subject is too long.